# The Chemical Signatures of Water Extract of *Zingiber officinale* Rosc

**DOI:** 10.3390/molecules27227818

**Published:** 2022-11-13

**Authors:** Fengying Lu, Hua Cai, Saimei Li, Wei Xie, Rongjin Sun

**Affiliations:** 1Department of Medical Genetics, Changzhou Maternal and Child Health Care Hospital, Changzhou Medical Center, Nanjing Medical University, Changzhou 213000, China; 2Hubei Key Laboratory of Wudang Local Chinese Medicine Research, School of Pharmaceutical Sciences, Taihe Hospital, Hubei University of Medicine, Shiyan 442000, China; 3Department of Shang Han Lun, Guangzhou University of Chinese Medicine, Guangzhou 510000, China; 4The First Affiliated Hospital of Guangzhou University of Chinese Medicine, Guangzhou 510000, China; 5Zhongshan Hospital of Traditional Chinese Medicine Affiliated to Guangzhou University of Chinese Medicine, Zhongshan 528400, China

**Keywords:** chromatography, ginger, mass spectrometry, plant extracts

## Abstract

Background: Ginger (*Z. officinale* Rosc.) is a common herb and is widely used as a diet-based or home therapy in traditional medicine worldwide. However, fresh ginger turns into dried ginger after kiln drying and shows a different treatment effect in clinical practice. Objective: To characterize the changes of major bioactive constituents in dried ginger after the processing of fresh ginger. Methods: A novel, ultra-high-performance liquid chromatography coupled with quadrupole time-of-flight tandem mass spectrometry (UHPLC–QTOF/MS) method was established to characterize the changes in the bioactive constituents of dried ginger. The novel strategy was split into two steps: firstly, the MS selected the most intense precursor ions of tandem MS; then, target MS/MS acquisition with different collision energies (10, 20, and 40 eV) was used to characterize the compound’s accurate MS/MS spectra and compare the MS/MS spectrum with the building MS reference library and reference standards. Result: Fifty-three compounds, including diarylheptanoids, gingerols, gingerodiols, gingerdiones, and shogaol-related compounds, were identified based on summarized fragmentation patterns. Fifteen out of fifty-three compounds were diarylheptanoids, which was different from fresh ginger. Conclusion: These identified compounds could be used to characterize the quality of dried ginger, pharmacologic studies should focus on diarylheptanoids explaining the different treatment effects between fresh ginger and dried ginger.

## 1. Introduction

Ginger (*Z.* Rosc.) is a common spice that is widely used as a diet-based or home therapy in various traditional systems of medicine around the world. It has been used in traditional medicine practice for the treatment of arthritis [1,2,3]; rheumatological diseases; gastrointestinal disorders such as distress symptoms, digestive disorders, and pain [4,5,6]. Clinical pharmacology research has shown that the active compounds in ginger are responsible for antioxidant and cancer-preventive properties [7,8,9], the prevention of chemotherapy-induced toxicity [10], and the improvement of inflammatory bowel disease and colitis [11,12]. However, the chemicals responsible for ginger’s pharmacology vary considerably.

Gingerols, shogaols, and their homologues are the major volatile components of *Zingiber officinale* Rosc. The percentage of volatile components varies based on the plantation region and harvesting season. Non-volatile components include paradols and their derivatives; zingerone; monoterpenoids; organic acids; and flavonoids [13,14,15,16,17]. In clinical practice, the decoction of dried ginger is a common practice in traditional medicine and shows different health effects from fresh ginger against chronic diseases [18]. One reason is that the major compounds in fresh ginger are liable to dehydrate and convert when exposed to heat and/or acidic conditions. The other is that during the decoction process, the active components vary sharply with changes in temperature and boiling time [19,20]. Therefore, in the current study, we explored the bioactive constituents in the aqueous extract of dried ginger.

The components of fresh ginger has been chiefly studied by high-performance liquid chromatographyor with a gas chromatography-mass spectrometer (GC-MS), because volatile oil is believed to be its main effective constituent [21]. These are also the usual ways to conduct quality assessments of ginger [22,23,24]. However, due to the characteristics of volatile oil, the quality control and bioactive components of ginger are still under investigation.

Mass spectrometry is currently one of the most robust and sensitive instrumental methods applied to the structural characterization of the secondary metabolite of ginger [25,26]. Therefore, we will use UHPLC-QTOF/MS to tentatively identify and characterize the chemical signature of the dried ginger. In addition, the gingerols and shogaols ’ homologues usually have the same skeletons as the parents. Therefore, the fragmentation of the parents molecules will provide specific structural information about the functionality of different compounds and is necessary and helpful for the characterization of the gingerols and shogaols ’ analogues in the tandem mass spectrometry (MS/MS) model.

## 2. Results and Discussion

### 2.1. LC/MS Conditions

In this study, tandem mass spectrometry (MS/MS) was used to identify the molecular ions of different components. The results were shown in Figure 1 and Figure 2 and Table 1. Most of the ingredients in dried ginger were identified within 40 min. Both positive and negative ionization modes were used to detect the compounds (Figure 1 and Figure 2). It was found that the signal in the QTOF/MS positive mode was much more sensitive than that in the negative mode. Different polarity compounds are detected in the two ionization modes. It looks like the gingerol and shogaol derivatives ionize better in the negative mode. In the tandem MS negative mode, the intensity of the compound fragments was too weak to analyze. About 90 different ions were presented in the positive MS mode. In total, we characterized 53 compounds.

### 2.2. Establishment of Fragmentation Patterns

Reference standards (zingerone, 6-gingerol, and 6-shogaol) were used in this study to demonstrate identity (Figure 3). These compounds were dissolved in 50% methanol at a final concentration of 1 μM for the UHPLC–ESI QTOF MS/MS analyses. Figure 3 shows the reference standards’ base peak chromatograms (BPC) in both positive and negative ionization modes. All constituents were identified by comparing the UHPLC retention time, accurate mass, and mass spectrum with those standards (Table 1). The abnormal peak in Figure 3 was the background contamination ion in the MS, and it did not impact our analysis.

### 2.3. Gingerol-Related Compounds

Gingerol-related compounds are the most important components in fresh ginger. Gingerol, gingerdione, gingerdiol, and shogaol contribute the predominant peppery taste in *Z. officinale Rosc*. Additionally, they all have a 4-hydroxy-3-methoxyphenylmoiety with different hydrocarbon chains. In our study, only a few gingerol-related compounds were identified in the extraction of dried ginger, including compound **33**([6]-gingerol), **41**(acetoxy-[6]-gingerol), **44**(acetoxy-[10]-gingerol), and **50** (dehydro-[8]-gingerol. These results are consistent with Tao and Li’s study [27].

### 2.4. Gingerdione-Related Compounds

Gingerdione, a gingerol derivative, is one of the major constituents of dried ginger. Compared to the standard compound, 6-gingerol, compound **25** showed a decrease of 4 Da for its corresponding [M+H]^+^ 291.1580 in (+) ESI-MS. Protonated ions of compound **25** were further fragmented by losing a neutral alkyl moiety and a rearrangement (Figure 4), leading to the formation of predomination A at *m*/*z* 177.0539 *m*/*z* 145.0227. Compounds **47** and **53** had similar fragmentation behaviors to compound **25**. Compared to compound **25**, the protonated ions of compounds **47** and **53** showed increases of 48 and 72 Da, respectively. Therefore, compounds **47** and **53** were tentatively identified as 1-dehydro-[10]-gingerdione and 1-dehydro-[12]-gingerdione.

### 2.5. Gingerdiol-Related Compounds

The precursor ion of compound **48** was sodium adduct ions in (+) ESI-MS, which is different from the standard compound, [6]-gingerol (Table 1). However, the fragmentation pattern of [6]-gingerdiol was similar to that of [6]-gingerol, and they both broke into *m*/*z* 177.0899 and 137.0589. According to the (+) ESI-MS/MS spectra for compound **43**, we know that the parent ion fragments into 321.2052[M+H–CH_3_COO]^+^ due to the loss of 60 Da (AcOH). This information suggests that the acetoxy group, and not the hydroxy group, is present on the aliphatic side chain of compound **48**. Compound **43** was identified as diacetoxy-[6]-gingerdiol. Compound **45** was identified as methyl-diacetoxy-[6]-gingerdiol. Sodium adduct ions (417.2227) fragmented into *m*/*z* 335.2210[M+H–CH_3_COO]^+^, 275.1911[M+H–2CH_3_COO]^+^, 177.0900, 151.0747, and 137.0588. Compounds **11** and **40** were identified as methyl-diacetoxy-[4]-gingerdiol and diacetoxy-[4]-gingerdiol, which had the same basic skeleton and fragmentation pathway as compound **45** and have been reported previously in the literature.

### 2.6. Shogaol and Paradol-Related Compounds

Compounds **13**, **42**, **34**, **49**, and **51** were identified as [4]-shogaol, [6]-shogaol, 6-hydroxy-[6]-shogaol, [10]-shogaol, and 6-hydroxy-[10]-shogaol. These are all shogaol derivatives and have similar fragmentation patterns according to the mass spectrum(B). This fragmentation pattern was consistent with Hongliang Jiang’s research results [28]. 6-hydroxy-[6]-shogaol and 6-hydroxy-[10]-shogaol can fragment into *m*/*z* 177.0535(A). Compounds **31** and **17** were identified as [6]-paradol and dihydro-[6]-paradol in positive ionization mode. They also fragmented into ions at *m*/*z* 137.0595—the same as the shogaol. [6]-paradol was detected as a protonated ion (*m*/*z* 279.0941) and fragmented into ions at *m*/*z* 261.1840[M+H–H_2_O]^+^, 233.0954[M+H–H_2_O–CO]^+^, and 137.0595(B).

### 2.7. Diarylheptanoids

More than 15 compounds were identified as diarylheptanoids, which are primarily responsible for cytotoxicity and apoptosis in ginger [9,29,30]. These diarylheptanoids were characterized by the presence of 5-hydroxy and 3-oxo groups on the heptane skeleton (Figure 5A,B).

Compounds **8**, **10**, **12**, **16**, **20**, and **37** are diarylheptanoids with -OH, -H, =O, or CH_3_CO atomic groups at the 1, 3, or 5 positions of the carbon chain. Compounds **18**, **21**, **24**, **30**, **32**, **35**, **36**, and **38** have CH_3_CO- at both the 3 and 5 positions of the carbon chain. Most of them can fragment into *m*/*z* 177.0690 (Figure 5A) or 137.0588 (Figure 5B) at different collision energies (Figure 6). We identified these compounds based on the characteristics of the compounds shown in research conducted by Riethmüller [31,32] and Svarc-Gajic [33]. The typical scheme of diarylheptanoid fragmentation was shown in Figure 6.

The precursor ions of compound **6** were observed at *m*/*z* 371.1481 [M+H]^+^ in (+) ESI-MS and *m*/*z* 369.1251 [M–H]^−^ in (–) ESI-MS, indicating a molecular weight of 370. When compared to compound **7**, compound **6** showed an increase of 2 Da on the precursor ion, and its product ions are shown in Table 1, indicating that it may be a homolog of compound **6** with a difference in the carbon–carbon double bond. The protonated ion compound **15** (387.1729) is 14 Da larger than compound **7** (373.1578), which indicates that compound **15** has a methoxy group. Compounds **9** and **14** displayed the same fragmentation behavior; however, protonated ion compound **14** (405.1885) is 14 Da larger than compound **9** (391.1731), differing in a methoxy group. The fragmentation pattern of compound **23** is unique and was identified as 3-acetoxy-5-hydroxy-1-(3,4-dihydroxy-5-methoxyphenyl)heptane by referring to the literature [34].

### 2.8. Two Amino Acids, Four Fatty Acids, and Others

The compound 1 deprotonated ion is 130.0870[M-H]^−^, and it fragments into *m*/*z* 115.0035[M–NH2]^−^ and 71.0140[M–NH_2_–COO]^−^ at the negative ionization mode with different collision energy. Therefore, compound 1 was identified as isoleucine. Compound **2** showed [M–H]^−^ ions with *m*/*z* 164.0713. The precursor ion and daughter ion matched with what we obtained from phenylalanine [34].

In the precursor ion scan spectrum, *m*/*z* of compound 3 [M+H]^+^ was 153.1282. It can fragment into *m*/*z* 93.0332[M+H–CH_2_COO]^+^ and 65.0383[M+H–CH_2_COO–CO]^+^ in the positive ionization mode. Therefore, compound 3 was identified as citral [35]. Compounds 28, 39, and 52 are fatty acids and were tentatively identified by comparison with the literature [36].

Compounds **4** and **29** were also identified in the extractions of *Z. mioga* and *Z. officinale*, and their daughter ions matched galanganol C and curcumadiol’s daughter ions [37].

## 3. Materials and Methods

### 3.1. Chemicals and Reagents

HPLC-grade acetonitrile, formic acid, and methanol were purchased from Merck KGaA (Darmstadt, Germany). Deionized water was re-distilled. Three standard materials—zingerone (Lot J0108AS), 6-gingerol (Lot O1014AS), and 6-shogaol (Lot O1020AS)—were purchased from Meilunbio company (Liaoning, Dalian, China, purity > 99.0%)

### 3.2. Plant-Material and Sample Preparation

Dried ginger materials appeared as yellow primrose pieces of wood and were purchased from Kangmei Chinese Traditional Medicine. Co. Ltd. (Guangdong, China). Original herbs were produced from Sichuan in 2017 (MAN: 2017.04.01)—10 g in each package (LOP: 170400541). The dried ginger was immersed in eight-fold volumes of water (1:8, w/v) for 30 min and boiled for 1 h. The decoction was filtered through 8 layers of gauze and was lyophilized to dried powder. We evaluated the amount of solvent (1:2, 1:4, and 1:8) and the extraction time (0.25, 0.5, 1, and 2 h) impact on the extraction efficiency, and we found that the ginger was boiled for a long time (2 h) or short time (0.5 h), removing valuable components from the extractions. Therefore, we closed 8-fold water and only boiled 1 h to acquire more peaks in the mass spectrum.

The LC-MS sample preparation: 1 mL of deionized water was added to 50 mg of the freeze-dried powder, vortexed for 10 min, and then sonicated for 15 min at 40 °C. The mixture was then vortexed vigorously for 3 min, after centrifugation at 15,000 rpm for 30 min. Five microliters of supernatant was injected into the UPLC–MS/MS system for qualitative analysis.

### 3.3. Accurate-Mass QTOF LC/MS System

The analysis of dried ginger was performed on an Agilent 1290UHPLC system coupled to an in-line diode array detector (DAD) and an Agilent 6540 Accurate-Mass QTOF LC/MS system with Agilent Jet Stream technology for electrospray ionization (Agilent Technologies, Santa Clara, CA, USA). The LC conditions were as follows: separation column, Acquity HPLC BEH C18 column (50 × 2.1 mm, i.d. 1.7 μM; Waters); the mobile phase consisted of 0.1% aqueous formic acid (*v*:*v*) (A) and acetonitrile (B), using a gradient elution of 5% B for 0–2 min, 5–10% B for 2–5 min, 10–14% B for 5–8 min, 14% B for 8–14 min, 14–17% B for 14–20 min, 17–20% B for 20–25 min, 20–25% B for 25–30 min, 25–50% B for 30–40 min, 50–100% B for 40–50 min, 100% B for 50–55 min, and 100–5% B for 55–60 min. The pastime was 3 min for the re-equilibrated systems. The flow rate was 0.5 mL/min, the temperature was 40 °C, and the injection volumes were 2 μL in MS mode and target MS/MS mode.

### 3.4. Mass Spectrometry

Full acquisition MS, auto MS/MS, and targeted MS/MS were performed with a 6540 QTOF Mass Spectrometer in both positive and negative ionization modes. Full acquisition MS spectra were collected over a mass range of *m*/*z* 100–1700, and the acquisition rate was 1 Hz at 1000 ms/scan. In the auto MS/MS and targeted MS/MS modes, the precursor MS spectrum was from *m*/*z* 100 to 1500, and the acquisition rate was 2 Hz with 500 ms/scan; the MS/MS spectrum was from *m*/*z* 50 to 1000, and the acquisition rate was 3 Hz with 333 ms/scan. The conditions of the ESI source were as follows: a drying gas (N2) flow rate of 11 (L/min); a drying gas temperature of 300 °C; a sheath gas temperature of 350 °C; a nebulizer at 40 psi; a capillary voltage of 3.5 kV (negative mode) or 4 kV (positive mode); a fragmentor at 175 V; a skimmer voltage of 60 V; and an octopole RF of 250 V. Every day, prior to the analyses of the samples, the mass axis was calibrated. All of the operations, acquisitions, and analyses of data were controlled by Mass Hunter software version B.06.00 (Agilent Technologies, Santa Clara, CA, USA).

### 3.5. Building the Chemical Database of Ginger

The ginger database was created using the Agilent software, Personal Compound Database Library (PCDL). The database contained the formulas, accurate masses, compound names, and original plants. The ginger database is available in Appendix A. The records of 236 compounds were input into the database by comprehensively searching databases such as Sci Finder, PubMed, TCM Database@Taiwan, Chinese National Knowledge Infrastructure of Tsinghua University, and KNApSAcK for all of the compounds reported in the literature for ginger (Appendix A).

### 3.6. Preparation of the Reference Standard

The stock solutions of the reference standard (gingerol, zingerone, and 6-shogaol) were prepared in ethanol/DMSO (4:1, *v*:*v*) at final concentrations of 10 mM. Then, each stock solution was diluted by 50% methanol to 1 μM for analysis.

## 4. Conclusions

A novel UHPLC–ESI–QTOF/MS approach was developed to identify chemical profiles of *Z. officinale* Rosc.. Many studies have demonstrated that the major function of the drying process is to reduce the gingerol concentration, increase the terpene hydrocarbon level, and convert some monoterpene alcohols into their corresponding acetates [13,24,38,39]. Our study results showed that the major components of dried ginger are diarylheptanoids. However, the main compounds in fresh ginger, gingerols, and shogaols, which are responsible for the bioactivity and spicy taste, were present in very low concentrations. This result may support the fact that dried ginger has different chemical constituents and pharmacological activities when compared with fresh herbs in clinical practice. Therefore, future pharmacologic studies should focus on these diarylheptanoids.

Most of the gingerol-related compounds have -OH and -OCH_3_ groups on their benzene ring. All of them fragment into the basement product ion *m*/*z* 137 (2-hydroxy-3-methoxyphenyl–CH_2_^+^ and 2,3-methoxyphenyl^+^). This fragmentation pattern was useful for diagnosing fragmentation behavior in positive and negative ESI–QTOF/MS, and analyzing the structures of homologs and allowed us to classify compounds by group and identify them based on key structural features. Overall, our novel strategy only requires 1–2 h to complete each peak, with each compound feature guided by the in-house database.

## Figures and Tables

**Figure 1 molecules-27-07818-f001:**
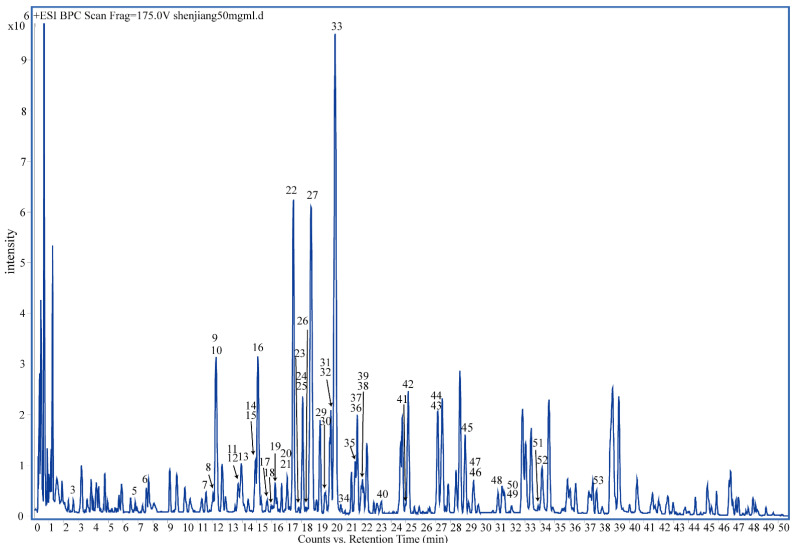
(+)-ESI base peak chromatogram (BPC) of the dried ginger water extract by UHPLC–ESI-QTOF-MS.

**Figure 2 molecules-27-07818-f002:**
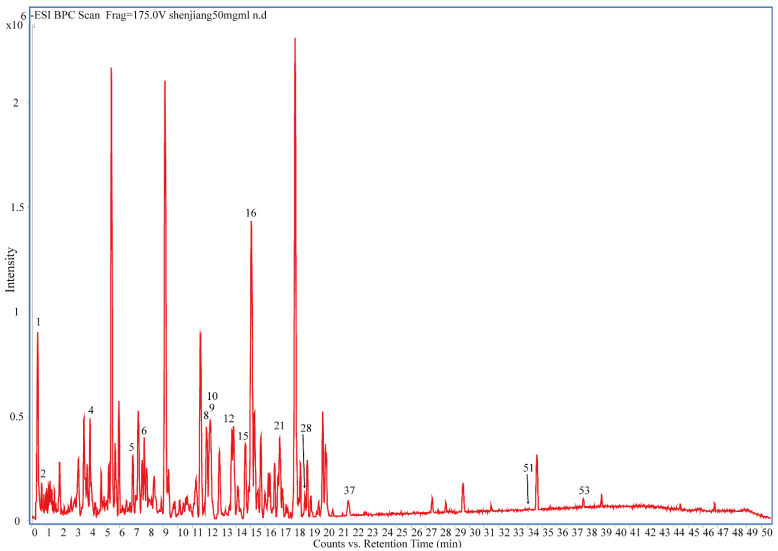
(−)-ESI base peak chromatogram (BPC) of the dried ginger water extract by UHPLC–ESI-QTOF-MS.

**Figure 3 molecules-27-07818-f003:**
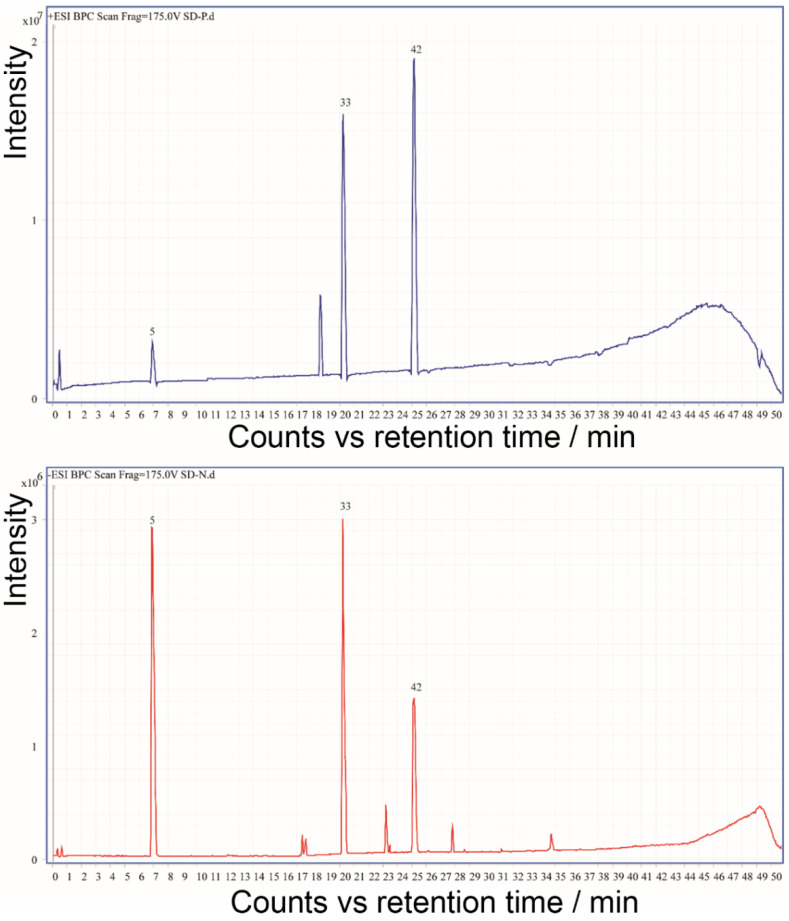
The base peak chromatograms (BPC) in positive negative and positive modes of the 3 standards of *Zingiber officinale* Rosc.: **5** zingerone, **33** 6-gingerol, and **42** 6-shogaol.

**Figure 4 molecules-27-07818-f004:**
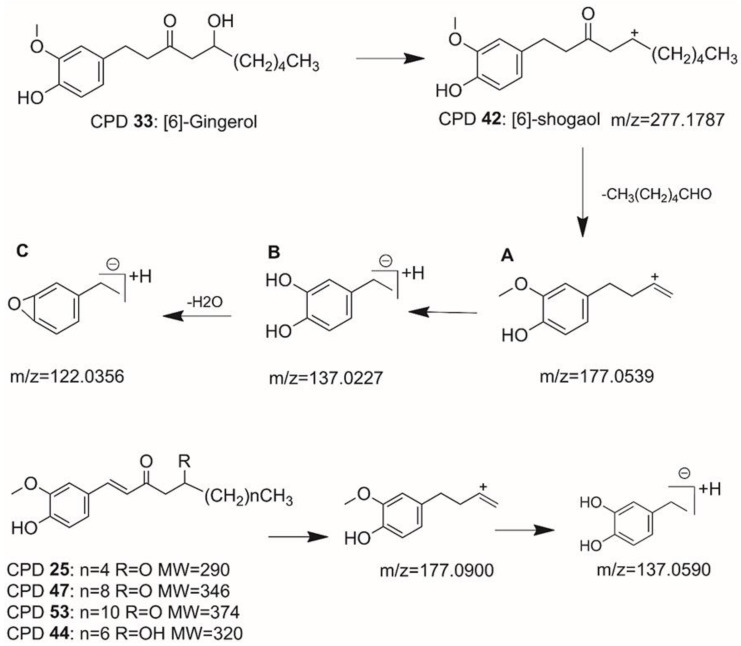
The typical fragmentation mechanisms of 6-gingerol and dehydro-gingerdione in positive mode. *m*/*z* 177.0900 (**A**), *m*/*z* 137.0590 (**B**), and *m*/*z* 122.0356 (**C**) were the base peaks of the derivative compounds in MS/MS.

**Figure 5 molecules-27-07818-f005:**
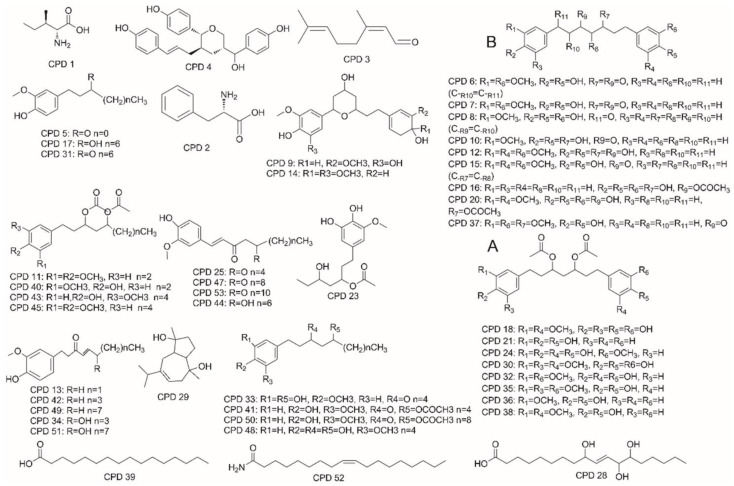
Chemical structure of the compounds identified by UHPLC–ESI–QTOF–MS/MS from dried ginger’s water extract. **A**, diarylheptanoids with two CH_3_CO atomic groups; **B** diarylheptanoids with one CH_3_CO atomic group.

**Figure 6 molecules-27-07818-f006:**
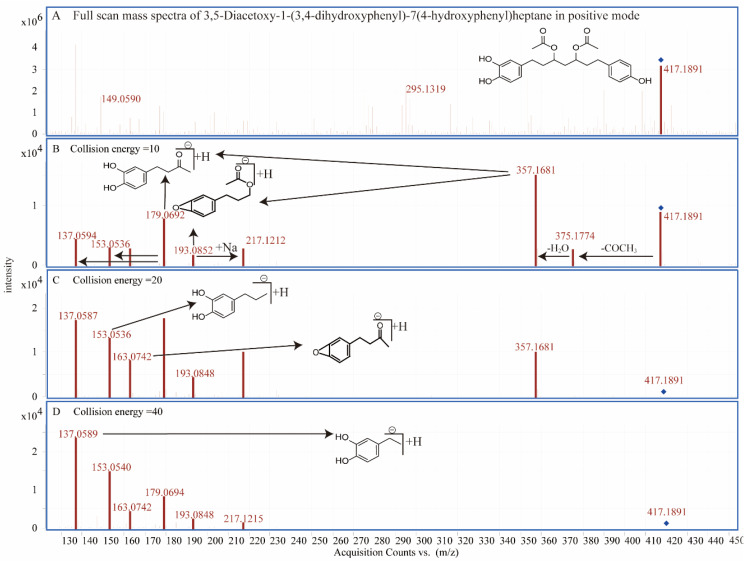
The mass spectra and the fragmentation pathway of compound 21, which was identified as 3,5-diacetoxy-1-(3,4-dihydroxyphenyl)-7(4-hydroxyphenyl) heptane. The mass spectra of compound 21 was shown in (**A**) with different collision energies 10 V (**B**), 20 V (**C**), and 40 V (**D**).

**Table 1 molecules-27-07818-t001:** UHPLC–ESI–QTOF–MS/MS results of the analysis of the dried ginger extract in the positive and negative models.

No.	Compound Name	Formula	Rt/ min	Detected Mass	Expected Tgt Mass	Diff/ ppm	Positive	Negative
							MS/MS	MS/MS
1	Isoleucine	C_6_H_13_NO_2_	0.50	131.0947	131.0946	0.14	N	130.0870[M–H]^−^, 115.0035[M–NH_2_]^−^, 71.0140[M–NH_2_–COO]^−^
2	Phenylalanine	C_9_H_11_NO_2_	0.69	165.0785	165.0790	−2.70	N	164.0713[M–H]^−^, 147.8934[M–H_2_O]^−^, 103.0557[M–H_2_O–COO]^−^, and 72.0098
3	(*Z*)-citral	C_10_H_16_O	2.66	152.1197	152.1201	−2.88	153.1282[M+H]^+^, 93.0332[M+H-CH2COO]^+^, and 65.0383[M+H–CH2COO–CO]^+^	N
4	Galanganol C	C_27_H_28_O_5_	4.07	432.1937	432.1937	12.83	N	431.1914[M–H]^−^, 389.1797, 179.0545, 89.0243
5	Zingerone *	C_11_H_14_O_3_	6.71	194.0942	194.0943	1.86	N	193.0863[M–H]^−^, 178.0573 [M–CH_3_–H]^−^, and 135.0450
6	Dihydrocurcumin	C_21_H_22_O_6_	7.75	370.1402	370.1416	−3.80	371.1481[M+H]^+^, 235.0943, 177.0535, 137.0589	369.1251[M–H]^−^
7	Tetrahydrocurcumin	C_21_H_24_O_6_	11.58	372.1557	372.1573	−4.18	373.1578[M+H]^+^, 179.0963, 153.0537, and 137.0589	
8	(*E*)-7-(3,4-dihydroxyphenyl)-1-(4-hydroxy-3-methoxyphenyl)hept-2-en-1-one	C_20_H_22_O_5_	12.00	342.1451	342.1467	−4.64	343.1525[M+H]^+^, 258.2470, 147.0438, 137.0586, 123.0431, 107.0481, and 86.0960	327.1564[M–H]^−^, 135.0441
9	1,5-epoxy-3-hydroxy-1-(3,4-dihydroxy-5-methoxyphenyl)-7-(4-hydroxy-3-methoxyphenyl)heptane	C_21_H_26_O_7_	12.18	390.1659	390.1679	−5.09	391.1731[M+H]^+^, 179.0689, and 137.0586	389.1616[M–H]^−^, 165.0554
10	5-hydroxy-1-(4-hydroxy-3-methoxyphenyl)-7-(4-hydroxyphenyl)-3-heptanone	C_20_H_24_O_5_	12.23	344.1599	344.1624	−7.13	345.1631[M+H]^+^, 258.1470, and 123.0431	343.1552[M–H]^−^
11	Methyl diacetoxy-[4]-gingerdiol	C_20_H_30_O_6_	13.73	366.2048	366.2042	1.39	389.1952[M+Na]^+^, 355.1524, 297.1104, 193.0484, and 137.0588	N
12	3,5-dihydroxy-1-(4-hydroxy-3,5-dimethoxyphenyl)-7-(4-hydroxy-3-methoxyphenyl)heptane	C_22_H_30_O_7_	13.73	406.1973	406.1992	−4.50	407.1992[M+H]^+^, 215.1382, 137.0584, 86.0959, and 70.0650	405.1530[M–H]^−^, 165.0558
13	[4]-Shogaol	C_15_H_20_O_3_	14.00	248.1403	248.1412	−3.84	249.1481[M+H]^+^, 163.0745, 137.0590, and 131.0480	N
14	1,5-epoxy-3-hydroxy-1-(4-hydroxy-3,5-dimethoxyphenyl)-7-(4-hydroxy-4-methoxyphenyl)heptane	C_22_H_28_O_7_	14.86	404.1819	404.1835	−3.90	405.1885[M+H]^+^, 217.1210, 167.0693, and 139.0854	N
15	Gingerenone B	C_22_H_26_O_6_	14.87	386.1714	386.1729	−4.01	387.1729[M+H]^+^, 247.1316, 193.0848, 167.0692, and 137.0592	385.1603[M–H]^−^, 341.1071, 223.0609, and 101.0248,
16	3-acetoxy-5-hydroxy-1-(4-hydroxyphenyl)-7-(3,4-dihydroxyphenyl)heptane	C_21_H_26_O_6_	15.04	374.1709	374.1729	−5.43	375.1780[M+H]^+^, 341.1720, 217.1207, 163.0744, and 137.0588	373.1661[M–H]^−^, 331.1540, and 175.0753
17	Dihydro-[6]-paradol	C_17_H_28_O_3_	15.64	280.2050	280.2038	4.13	303.1941[M+Na]^+^, 287.1989, 163.0742, and 103.0383	N
18	3,5-diacetoxy-1,7-bis(3,4-dihydroxy-5-methoxyphenyl)heptane	C_25_H_32_O_10_	15.96	492.1971	492.1996	−5.00	510.2883, 235.1176, 137.0589, 110.0708	N
19	Isomer of number 18	C_25_H_32_O_10_	16.10	492.1973	492.1996	−4.41	510.2883, 235.1176, 137.0589, and 110.0708	N
20	3-acetoxy-5-hydroxy-1-(3,4-dihydroxy-5-methoxyphenyl)-7-(4-hydroxy-3-methoxyphenyl)heptane	C_23_H_30_O_8_	17.01	434.1924	434.1941	−3.89	435.1989[M+H]^+^, 385.1987, 357.1692, 207.1003, 193.0846,181.0847,167.0694,163.0745,153.0539, and 137.0590	N
21	3,5-diacetoxy-1-(3,4-dihydroxyphenyl)-7(4-hydroxyphenyl)heptane	C_23_H_28_O_7_	17.01	416.1817	416.1835	−4.37	417.1890[M+H]^+^, 324.1414, 217.1211, 207.1009, 153.0539, 137.0590, and 81.0694	415.1761[M-H]^−^, 371.2049, and 359.1597
22	Isomer of number 45	C_22_H_34_O_6_	17.45	394.2334	394.2355	N	N	N
23	3-acetoxy-5-hydroxy-1-(3,4-dihydroxy-5-methoxyphenyl)heptane	C_24_H_32_O_8_	17.97	448.2075	448.2097	−4.93	449.2141[M+H]^+^, 373.1634, 313.1423,123.0435,	N
24	3,5-diacetoxy-1-(3,4-dihydroxyphenyl)-7-(3,4-dihydroxy-5-methoxyphenyl)heptane	C_24_H_30_O_9_	18.06	462.1860	462.1890	−6.41	480.2202, 179.0685, and 137.0593	N
25	Dehydro-[6]-gingerdione	C_17_H_22_O_4_	18.07	290.1526	290.1518	2.86	177.0538, 145.0277	N
26	Isomer of number 18	C_25_H_32_O_10_	18.38	492.1969	492.1996	−4.38	510.2883, 235.1176, 137.0589, and 110.0708	N
27	unknown	unknown	18.71	466.2413	466.2413	2.34	Unknown	N
28	Trihydroxy octadecenoic acid	C_18_H_34_O_5_	18.72	330.2405	330.2406	−0.40	N	329.233[M-H]^−^, 283.2613
29	Curcumadiol	C_15_H_26_O_2_	19.24	238.1946	238.1933	7.94	261.1839[M+Na]^+^, 229.1566, 177.0886, 163.0743, 145.0637, 137.0590, 131.0488, 117.0692, and 103.0536	N
30	3,5-diacetoxy-1-(3,4-dihydroxy-5-methoxyphenyl)-7-(4-hydroxy-3,5-dimethoxyphenyl)heptane	C_26_H_34_O_10_	19.64	506.2129	506.2152	−4.61	507.2168[M+H]^+^, 355.1523, 215.1054, 179.0695, and 137.0587	N
31	[6]-Paradol	C_17_H_26_O_3_	19.91	278.1869	278.1882	−4.74	279.0941[M+H]^+^, 261.1840[M+H–H_2_O]^+^, 233.0954[M+H–H_2_O–CO]^+^, and 137.0595	N
32	3,5-diacetoxy-1-(4-hydroxy-3-methoxyphenyl)-7-(3, 4-dihydroxy-5-methoxyphenyl)heptane	C_25_H_32_O_9_	19.93	476.2032	476.2046	−2.94	477.2855[M+H]^+^, 285.2163, 179.0695, 137.0591, and 69.0693	N
33	[6]-Gingerol*	C_17_H_26_O_4_	20.24	294.1819	294.1831	−4.10	317.1712[M+Na]^+^, 177.0906, 137.0600, 99.0797, 69.0697	N
34	6-hydroxy-[6]-shogaol	C_17_H_24_O_4_	20.63	292.1664	292.1675	−3.79	293.1742[M+H]^+^, 179.0690, and 137.0588	N
35	3,5-diacetoxy-1-(4-hydroxy-3,5-dimethoxyphenyl)-7-(4-hydroxy-3-methoxyphenyl)heptane	C_26_H_34_O_9_	21.60	490.2176	490.2203	−5.53	491.2525[M+H]^+^, 431.2055[M+H–CH_3_COO]^+^, 371.1837[M+H–2CH_3_COO]^+^, 339.1577, 247.1314, and 193.0852	N
36	3,5-diacetoxy-1-(4-hydroxy-3-methoxyphenyl)-7-(4-hydroxyphenyl)heptane	C_24_H_30_O_7_	21.74	430.1969	430.1992	−5.31	431.2926[M+H]^+^, 193.0848, and 167.0691	N
37	1,7bis-(4-hydroxy-3-methoxyphenyl)-5-methoxy-3-heptanone	C_22_H_28_O_6_	21.83	388.1887	388.1886	0.15	N	387.1814[M–H]^−^, 329.1384, 207.1025, 165.0552, and 122.0372
38	3,5-diacetoxy-1,7-bis(4-hydroxy-3-methoxyphenyl)heptane	C_25_H_32_O_8_	22.01	460.2078	460.2097	−4.10	478.2418[M+NH_4_]^+^, 341.1732, 217.1212, and 137.0593	N
39	Palmitic acid	C_16_H_32_O_2_	22.16	256.2392	256.2402	−4.03	257.2620[M+H]^+^, 191.1054, and 106.0856	N
40	Diacetoxy-[4]-gingerdiol	C_19_H_28_O_6_	23.35	352.1863	352.1886	−6.48	370.2203[M+H_2_O]^+^, 137.0591	N
41	Acetoxy-[6]-gingerol	C_19_H_28_O_5_	25.12	336.1921	336.1937	−4.82	337.1985[M+H]^+^, 279.0995, 261.0898, 163.0739, 137.0590, 131.0484, 122.0359, 103.0537, and 94.0409	N
42	[6]-shogaol*	C_17_H_24_O_3_	25.24	276.1719	276.1725	−4.19	277.1787[M+H]^+^, 137.0591, 122.0356, and 94.0408	N
43	Diacetoxy-[6]-gingerdiol	C_21_H_32_O_6_	27.14	380.2178	380.2199	−5.60	403.2071[M+Na]^+^, 321.2052 [M+H-CH3COO]^+^, 137.0592	N
44	Dehydro-[8]-gingerol	C_19_H_28_O_4_	27.14	320.1975	320.1988	−3.93	321.2049[M+H]^+^, 261.1840, 177.0904,163.0742, and 137.0590	N
45	Methyl,diacetoxy-[6]-gingerdiol	C_22_H_34_O_6_	28.96	394.2334	394.2355	−5.39	417.2227[M+Na]^+^, 412.2678[M+H2O]^+^, 335.2210[M+H–CH_3_COO]^+^, 275.1911[M+H–2CH_3_COO]^+^, 177.0900, 151.0747, 137.0588,	N
46	Isomer of number 21	C_23_H_28_O_7_	29.40	416.1817	416.1835	N	417.1890[M+H]^+^, 324.1414, 217.1211, 207.1009, 153.0539, and 137.0590	N
47	Dehydro-10-gingerdione	C_21_H_30_O_4_	29.55	346.2141	346.2144	−0.82	347.2206[M+H]^+^, 177.0902, and 137.0592	N
48	[6]-gingerdiol	C_17_H_28_O_4_	31.20	296.1997	296.1988	3.27	319.1890[M+Na]^+^, 177.0899, 137.0589, and 94.0399	N
49	[10]-shogaol	C_21_H_32_O_3_	32.11	332.2334	332.2351	−5.14	333.2410[M+H]^+^, 137.0590, 94.0403	N
50	Acetoxy-[10]-gingerol	C_23_H_36_O_5_	32.11	392.2537	392.2563	−6.59	393.2606[M+H]^+^, 195.1220, 163.0737, and 137.0592	N
51	6-hydroxy-[10]-shogaol	C_21_H_32_O_4_	34.06	348.2284	348.2301	−4.88	349.2357[M+H]^+^, 179.0696, 161.0949, 137.0588, 121.0587, 95.0847	347.2227[M–H]^−^
52	Oleamide	C_18_H_35_NO	34.16	281.2708	281.2719	−3.82	282.2781[M+H]^+^, and 187.0727	N
53	Dehydro-[12]-gingerdione	C_23_H_34_O_4_	37.81	374.2440	374.2457	−4.52	375.2514[M+H]^+^, 177.0901, and 137.0590	373.1646[M–H]^−^, 313.1437, 191.1078, and 122.0374

* means identified by Standard reference. N means not detected, Rt means retention time.

## Data Availability

All data used to support the findings of this study are available from the corresponding author upon request.

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
