# Peer review of "The Chemical Signatures of Water Extract of Zingiber officinale Rosc"

_molecules, 2022, doi:10.3390/molecules27227818_

Round 1

Reviewer 1 Report

The article by Sun et al. deals with the development of an UHPLC-ESI-QTOF/MS approach to identify chemical profiles of Zingiber Officinale Roscoe.

In my opinion, the investigation has been carried out in a very basic and somehow confused form, and it add nothing to what it already known on the matter: the analytical method proposed by the authors for ginger is not new (see, for example, the 2007 article entitled “Characterization and identification of diarylheptanoids in ginger (Zingiber officinale Rosc.) using high-performance liquid chromatography/electrospray ionization mass spectrometry”, ref. 27 of the present manuscript) and none compound presented in the present manuscript has been identified for the first time (see, for example, the 2009 article entitled: “Identification and Quantification of Gingerols and Related Compounds in Ginger Dietary Supplements Using High Performance Liquid Chromatography-Tandem Mass Spectrometry”, ref. 25 of the present manuscript). More recently, a similar article entitled “Diarylheptanoid analogues from the rhizomes of Zingiber officinale and their anti-tumour activity” was published in RSC Adv., 2021, 11, 29376-29384, that the authors did not quote. Finally, the effect of heat on ginger composition is already known (see, for example, the 2016 article entitled: “Chemical characterization and antioxidant activities comparison in fresh, dried, stir-frying and carbonized ginger”).

Furthermore, the chromatographic method proposed experienced an instrumental failure (see, for example, the deformed chromatographic peak shapes showed in Figure 3).

For these reasons, I do not recommend this article for publication in Molecules.

Author Response

Reviewer 1:

Thanks so much for the valuable comments on our manuscript. The reviewer critically evaluated the innovation and the significance of our manuscript. But I want to ask the reviewer to rethink the issue of our manuscript after reading my clarification.

  1. The article by Sun et al. deals with the development of an UHPLC-ESI-QTOF/MS approach to identify chemical profiles of Zingiber Officinale Roscoe.

In my opinion, the investigation has been carried out in a very basic and somehow confused form, and it add nothing to what it already known on the matter: the analytical method proposed by the authors for ginger is not new (see, for example, the 2007 article entitled “Characterization and identification of diarylheptanoids in ginger (Zingiber officinale Rosc.) using high-performance liquid chromatography/electrospray ionization mass spectrometry”, ref. 27 of the present manuscript) and none compound presented in the present manuscript has been identified for the first time (see, for example, the 2009 article entitled: “Identification and Quantification of Gingerols and Related Compounds in Ginger Dietary Supplements Using High-Performance Liquid Chromatography-Tandem Mass Spectrometry”, ref. 25 of the present manuscript). More recently, a similar article entitled “Diarylheptanoid analogues from the rhizomes of Zingiber officinale and their anti-tumour activity” was published in RSC Adv., 2021, 11, 29376-29384, that the authors did not quote. Finally, the effect of heat on ginger composition is already known (see, for example, the 2016 article entitled: “Chemical characterization and antioxidant activities comparison in fresh, dried, stir-frying and carbonized ginger”).

1.Response to the reviewer :

In references 25 and 27 of the present manuscript, the authors identified and quantified the Gingerols and Related Compounds by LC-MS. The analytical method is different from ours. QTOF mass spectrometers are shown to be excellent devices for qualitative screening and identification. On the other hand, the LC-MS and QQQ mass spectrometers are excellent devices for quantitative targeted confirmation(see the link below).5990-3450EN_5989_5672.qxd (agilent.com).

The paper titled  “Diarylheptanoid analogues from the rhizomes of Zingiber officinale and their anti-tumour activity” is a very classical research paper for natural product chemistry, including Isolation, Identification and Biological Activity. They identified eight previously undescribed diarylheptanoids. The analytical method and research strategy are different from ours, too. We will cite this paper to support our results.

For our method, we can’t accurately identify new compounds in the ginger (Zingiber officinale Rosc.) using QTOF(also need others UV, NMR, and IR spectrum ), We only can tentatively identify the compounds based on the accurate mass (MS1), acquire MS2 spectra and comparing with the reference standard. The reason why we chose QTOF instruments is that they are well suited for untargeted metabolomics and they can acquire MS1 data over a wide m/z range in a single scan, thereby providing a broad survey of chemicals present in a sample. Additionally, QTOFs provide accurate mass, high resolving power, and the potential to acquire MS2 spectra, all of which are required for the confident identification of an unknown compound. 

So, this is our advantage. We developed a novel strategy to characterize the changes of major bioactive constituents in dried ginger.The novel strategy was split into two steps: firstly, the MS selected the most intense precursor ions of tandem MS, then target MS/MS acquisition with different collision energy (10, 20, and 40 eV ) was used to characterize the compound’s accurate MS/MS spectra, and compare the MS/MS spectrum with the building MS reference library and reference standards(please see the abstract).  This strategy will require 1-2 hours to complete each peak, termed a compounds feature Guided by the in-house Database(please see supplement of the in-house database ).  In addition, this method will be easily repeated in other labs without any technical issues. We added this part to the discussion.

Therefore, if we evaluate the manuscripts from the scope/angle of analytical chemistry, the innovation and the significance of our manuscript should be ok.

  1. Furthermore, the chromatographic method proposed experienced an instrumental failure (see, for example, the deformed chromatographic peak shapes showed in Figure 3).

Response to the reviewer :

For the method, I published my first LC-MS assay in 2015 and continued to work in the area of bioanalysis chemistry for 7 years (the link for my Bibliography: https://www.ncbi.nlm.nih.gov/myncbi/1zIpflsKA4yw9k/bibliography/public/).

I think I can explain the abnormal chromatographic peak in figure 3 according to my experience. There are  3 reference compounds in the samples, therefore it should show 3 peaks in the chromatograms. However, there were 4 and 9 peaks in the positive and negative model chromatograms, respectively.

The reason is the background contamination ion, common background contamination ions encountered in QTOF are polyethylene glycol, polypropylene glycol, phthalates, organic solvent clusters, solvent modifiers, fatty acids, metal ions, tritons, tweens and siloxanes, and its hard to clean them.  However, it didn’t impact our analysis in the positive model. We added one sentence to clarify the abnormal peak: The abnormal peak in figure 3 was the background contamination ion in the MS, and it didn’t impact our analysis.

Reviewer 2 Report

The manuscript “The Chemical Signatures of Water Extract of Zingiber Officinale Rosc.” authored by Fengyin Lu, Hua Cai, Saimei Li, Wei Xie, and Rongjin Sun, characterize the changes of major bioactive constituents between fresh and dried ginger by UHPLC-Q-TOF/MS. The comments and suggestions for the manuscript are as the following.

1.     Zingiber officinale should be revised as Z. offincinale, except for the time.

2.     Page 2 Line 76. A space insert between 10 and g.

3.     Page 4. The resolution of Figure 1 is very poor and the labeling number to peak above is very unclear. Please revise.

4.     Page 4. No peak in number 51. Please check.

5.     Page 5. In the Table 1, many chemical name of compounds should be revised carefully. Such as (Z)-citral (Z)-citral; (E)-7-(3,4-dihydroxyphenyl)-1-(4-hydroxy-3-methoxyphenyl)hept-2-en-1-one (E)-7-(3,4-dihydroxyphenyl)-1-(4-hydroxy-3-methoxyphenyl)hept-2-en-1-one; [M+H-CH2COO]+, 65.0383[M+H–CH2COO–CO]+ [M+H-CH2COO]+, 65.0383 [M+H–CH2COO–CO]+……etc.

6.     Page 12 Line 12. “including compound 33([6]-gingerol), compound 41(acetoxy-[6]-gingerol), 44(acetoxy-[10]-gingerol), and 50 (dehydro-[8]-gingerol.” “including compound 33 ([6]-gingerol), 41 (acetoxy-[6]-gingerol), 44 (acetoxy-[10]-gingerol), and 50 (dehydro-[8]-gingerol.)”

7.     Page 12 Line 12. A space insert between 4 and Da.

8.     Page 12 Line 37. A space insert between 60 and Da.

9.     Page 13 Line 41. “m/z 335.2210[M+H–CH3COO]+, 41275.1911[M+H–2CH3COO]+, 177.0900, 151.0747, 137.0588.” should be revised as “m/z 335.2210 [M+H–CH3COO]+, 41275.1911 [M+H–2CH3COO]+, 177.0900, 151.0747, and 137.0588.”

10.  Page 13 Line 42. “Compounds 11, and 40 “Compounds 11 and 40

11.  Page 13 Line 47. “Compounds13, 42, 34, 49,and 51 “Compounds 13, 42, 34, 49, and 51

12.  Page 13 Line 61. “Compounds 8, 10, 12, 16, 20, and 37 are diarylheptanoids with -OH, -H, =O or CH3CO atomic groups” “Compounds 8, 10, 12, 16, 20, and 37 are diarylheptanoids with -OH, -H, =O or CH3CO atomic groups”

13.  Page 13 Line 47. “Compounds 18, 21, 24, 30, 32, 35, 36, and 38 have CH3CO-“ “Compounds 18, 21, 24, 30, 32, 35, 36, and 38 have CH3CO-“

14.  Page 15, Line 80. “Compounds9 and 14 “Compounds 9 and 14

15.  Page 15, Line 83. A space insert into 23 and is.

16.  Page 15, Line 87. “115.0035[M–NH2] and 71.0140[M–NH2–COO]-“115.0035 [M–NH2] and 71.0140 [M–NH2–COO]-“.

17.  Page 15, Line 96. Zingiber mioga and officinale should be italic.

18.  Page 15, Line 85. The paragraph should be merged into paragraph 3.8 without italics.

19.  References part. Many scientific name don’t be italic. The format of references must follow the guideline of Molecules.

Author Response

Authors Response

Thanks so much for the valuable comments, which help us to improve the quality of our work. In the document, all changes were highlighted in red and the Track Changes is on.

Please check the point-to-point response as follows:

Comment

Response

1. Zingiber officinale should be revised as Z. offincinale, except for the time.

Modified as requested

2. Page 2 Line 76. A space insert between 10 and g.

Modified as requested

3. Page 4. The resolution of Figure 1 is very poor and the labeling number to peak above is very unclear. Please revise.

We improved the resolution of all the figures to 300DPI

4. Page 4. No peak in number 51. Please check.

Modified as requested

5. Page 5. In the Table 1, many chemical name of compounds should be revised carefully. Such as (Z)-citral →(Z)-citral; (E)-7-(3,4-dihydroxyphenyl)-1-(4-hydroxy-3-methoxyphenyl)hept-2-en-1-one → (E)-7-(3,4-dihydroxyphenyl)-1-(4-hydroxy-3-methoxyphenyl)hept-2-en-1-one; [M+H-CH2COO]+, 65.0383[M+H–CH2COO–CO]+ → [M+H-CH2COO]+, 65.0383 [M+H–CH2COO–CO]+……etc.

Modified as requested, and edited all the spelling issues.

6. Page 12 Line 12. “including compound 33([6]-gingerol), compound 41(acetoxy-[6]-gingerol), 44(acetoxy-[10]-gingerol), and 50 (dehydro-[8]-gingerol.” →“including compound 33 ([6]-gingerol), 41 (acetoxy-[6]-gingerol), 44 (acetoxy-[10]-gingerol), and 50 (dehydro-[8]-gingerol.)”

Modified as requested

7. Page 12 Line 12. A space insert between 4 and Da.

Modified as requested

8. Page 12 Line 37. A space insert between 60 and Da.

Modified as requested

9. Page 13 Line 41. “m/z 335.2210[M+H–CH3COO]+, 41275.1911[M+H–2CH3COO]+, 177.0900, 151.0747, 137.0588.” should be revised as “m/z 335.2210 [M+H–CH3COO]+, 41275.1911 [M+H–2CH3COO]+, 177.0900, 151.0747, and 137.0588.”

Modified as requested

10. Page 13 Line 42. “Compounds 11, and 40” →  “Compounds 11 and 40”

Modified as requested

11. Page 13 Line 47. “Compounds13, 42, 34, 49,and 51” →  “Compounds 13, 42, 34, 49, and 51”

Modified as requested

12. Page 13 Line 61. “Compounds 8, 10, 12, 16, 20, and 37 are diarylheptanoids with -OH, -H, =O or CH3CO atomic groups” → “Compounds 8, 10, 12, 16, 20, and 37 are diarylheptanoids with -OH, -H, =O or CH3CO atomic groups”

Modified as requested

13. Page 13 Line 47. “Compounds 18, 21, 24, 30, 32, 35, 36, and 38 have CH3CO-“ →“Compounds 18, 21, 24, 30, 32, 35, 36, and 38 have CH3CO-“

Modified as requested

14. Page 15, Line 80. “Compounds9 and 14” → “Compounds 9 and 14”

Modified as requested

15. Page 15, Line 83. A space insert into 23 and is.

Modified as requested

16. Page 15, Line 87. “115.0035[M–NH2]– and 71.0140[M–NH2–COO]-“ → “115.0035 [M–NH2] and 71.0140 [M–NH2–COO]-“.

Modified as requested

17. Page 15, Line 96. Zingiber mioga and officinale should be italic.

Modified as requested

18. Page 15, Line 85. The paragraph should be merged into paragraph 3.8 without italics.

Modified as requested

19. References part. Many scientific name don’t be italic. The format of references must follow the guideline of Molecules.

Modified the references according to the guideline of Molecules

Reviewer 3 Report

Please delete underlined characters, in the text.

Please check the characters at paragraph 3.9.

Please check the plant name in the discussion section (line 100).

How was the extraction condition optimized?

Please check the form of table 1.

Please format the reference section according the journal guidelines.

Please include missing sections, among which authors' contrubutions, supplementary materials...

Author Response

Authors Response

Thanks so much for the valuable comments, which help us to improve the quality of our work. In the document, all changes were highlighted in red and the Track Changes is on.

Please check the point-to-point response as follows:

Comment

Response

Please delete underlined characters, in the text.

Modified as requested

Please check the characters at paragraph 3.9.

Modified as requested

Please check the plant name in the discussion section (line 100).

Modified as requested

How was the extraction condition optimized?

We described the optimization as follows:

We evaluated the amount of solvent (1:2, 1:4, and 1:8) and extraction time (0.25, 0.5, 1 and 2hrs) impact on the extraction efficiency, we found that the ginger was boiled for long-time 2hrs or short-time 0.5hrs removing valuable components from the extractions. Therefore, we closed 8-fold water and only boiled 1 hour to acquire more peaks in the mass spectrum.

Please check the form of table 1.

Modified as requested

Please format the reference section according the journal guidelines.

Modified the references according to the guideline of Molecules

Please include missing sections, among which authors' contributions, supplementary materials...

Modified as requested, we added the following sections in the manuscript:

Authors' contributions: Hua Cai and Fengying Lu wrote the initial manuscript with major input from Wei Xie and Rongjin Sun. Hua Cai, Fengying Lu, Wei Xie performed experiments and analyzed the results, Saimei Li did the data interpretation. Saimei Li, Wei Xie and Rongjin Sun designed the study and obtained funding for the studies. All authors revised and approved the final manuscript.

Conflicts of Interest: The authors declare no conflict of interest.

Data Availability Statement: Yes.

Sample Availability: Dried Ginger are available from the authors.

Round 2

Reviewer 1 Report

My opinion on this article did not change, the main concern being the fact that it adds nothing to what it is already known on the matter: the method proposed is not new and the compounds identified are not new; even the use of the HPLC-ESI-Q-TOF-MS/MS has been already investigated for fresh ginger, as reported in reference 37.

For these reasons, I do not recommend this article for publication in Molecules.